# On the Expressiveness of Softmax Attention: A Recurrent Neural Network Perspective

**Gabriel Mongaras**                                                                          *gabriel@mongaras.com*
*Department of Computer Science*
*Southern Methodist University*

**Eric C. Larson**                                                                          *eclarson@smu.edu*
*Institute for Computational Biosciences*
*Southern Methodist University*

**Reviewed on OpenReview:** *https: // openreview. net/ forum? id= PHcITOi3vV*

## Abstract

Since its introduction, softmax attention has become the backbone of modern transformer architectures due to its expressiveness and scalability across a wide range of tasks. However, the main drawback of softmax attention is the quadratic memory requirement and computational complexity with respect to the sequence length. By replacing the softmax nonlinearity, linear attention and similar methods have been introduced to avoid the quadratic bottleneck of softmax attention. Despite these linear forms of attention being derived from the original softmax formulation, they typically lag in terms of downstream accuracy. While strong intuition of the softmax nonlinearity on the query and key inner product suggests that it has desirable properties compared to other nonlinearities, the question of why this discrepancy exists still remains unanswered. This work demonstrates that linear attention is a first-order approximation of the softmax numerator by deriving its full recurrent form. We further show empirically that the denominator's function can be effectively replaced by a simple vector norm. Using this form, each part of softmax attention can be described in the language of recurrent neural networks (RNNs). Describing softmax attention as an RNN allows for the ablation of the components of softmax attention to understand the importance of each part and how they interact. In this way, our work helps explain why softmax attention is more expressive than its counterparts.[1]

## 1 Introduction and Background

The formulation of softmax attention was proposed by Bahdanau et al. (2015) as a weighting mechanism for aligning recurrent neural networks (RNNs) in encoder-decoder architectures for language translation Sutskever et al. (2014). However, the modern day usage of the attention in a transformer architecture was first employed by Vaswani et al. (2017) as a way to do sequence mixing within the context of language translation. This formulation used the softmax activation function to model token routing without the use of a traditional recurrent network. Since its introduction, the attention mechanism has been widely adopted in various domains such as computer vision Dosovitskiy et al. (2021), generative models Esser et al. (2024), timeseries analysis Nie et al. (2023), audio processing Baevski et al. (2020), graphs Veličković et al. (2018), and many more applications.

While softmax attention is a powerful mechanism that is used in a variety of areas, a major drawback is its quadratic complexity with respect to the sequence length, $N$. Linear attention swaps the softmax nonlinearity

---

[1]Code found at: https://github.com/gmongaras/On-the-Expressiveness-of-Softmax-Attention-A-Recurrent-Neural-Network-Perspective

with a decomposable kernel function, reducing the complexity from quadratic to linear with respect to the sequence length. The original introduction of linear attention Katharopoulos et al. (2020) replaced the nonlinearity with an $elu(x) + 1$ kernel. Since the original formulation, other methods have been developed that replace the nonlinearity on the QK-softmax inner product with various other functions such as ReLU Xie et al. (2025), cosine similarity Mongaras et al. (2025), cosine reweighting Qin et al. (2022), or by decomposing the QK-softmax inner product Wang et al. (2020); Zhuoran et al. (2021). Although these methods are linear in complexity, none are as performant as softmax attention in terms of downstream accuracy.

In this work, we propose a recurrent reformulation of softmax attention and use this to describe the elements of softmax attention that make it more performant compared to its linear and sparse counterparts. While softmax attention is the driving method used in many architectures, the question of why the discrepancy between softmax attention and linear attention exists remains unanswered. This work provides a principled analysis of softmax attention by deriving a recurrent formulation using its Taylor series expansion. Through this formulation, we experiment to reveal how each recurrent component contributes to downstream accuracy using targeted ablations. We demonstrate that softmax attention is not merely a heuristic construction but a structured process with interpretable, sequential dynamics. In doing so, we bridge the gap between the observed empirical performance for linearized attention (and its variants) and the theoretical underpinnings of softmax attention.

## 2 Related Work

**Expressiveness of Linear Attention**  Many approaches have attempted to make linear attention more expressive to match softmax attention, while still retaining linear complexity. Choromanski et al. (2021) used linear approximations of the softmax kernel to achieve more performant linear attention mechanisms. As linear attention can be interpreted as an RNN Katharopoulos et al. (2020), Peng et al. (2025) proposed receptance weighted key values (RWKV) attempting to enhance the expressivity of RNNs with a receptance vector for time mixing. While this method helped to address computational complexity of RNN training, it still relied on linear attention in its formulation. Mamba Dao and Gu (2024) employed state space models Gu et al. (2022) to develop an efficient and expressive form of linear attention. Sun et al. (2025) formulated linear attention as a step in gradient decent of a hidden view. Another approach treats modeling the hidden state as a step in gradient decent Sun et al. (2025) which can be viewed as a type of linear attention. Behrouz et al. (2024) proposed Titans, which built upon this gradient formulation creating different variants of the hidden view gradient descent. ATLAS Behrouz et al. (2025) develops a new kind of recurrent models, leveraging test-time compute as in Behrouz et al. (2024). They include a proof show that recurrent models using higher order hidden states are more expressive. Sieber et al. (2024) makes a similar derivation as this work, but in the context of control systems, but did not apply this derivation to a recurrent architecture. Nauen et al. (2024) makes a simple derivation, but only explores second-order models. Although all of these methods are more expressive than linear attention, they are difficult to implement efficiently in hardware, akin to RNNs, and fall short to the accuracy of softmax attention.

**Softmax Attention Improvements**  Another direction of research explores how to improve softmax attention. A number of works attempt to improve softmax attention by increasing sparsity to reduce dependence on context length such as Longformer Beltagy et al. (2020), BigBird Zaheer et al. (2020), and randomized feature attention Peng et al. (2021). RoFormer Su et al. (2024) adds relative positional encodings to improve long sequence modeling and is used in most modern transformer models. Zhai* et al. (2023) prevents attention score entropy collapse by reparameterizing the weight matrices, modern transformers use norms to fix this issue. Newer improvements such as the Forgetting Transformer Lin et al. (2025) and DeepSeek Liu et al. (2024) make changes to the input of the attention mechanism such as combining the key and value matrices to reduce the KV cache size, or adding a "forget" gate. Many of these improvements make softmax attention slightly less computational, less memory intensive, or more expressive. However, the core mechanism driving the modeling capability is still softmax attention.

**Why Softmax?**  There are several works that also attempt to address the question: What makes softmax attention so performant? This fundamental question has eluded researchers for some time. For example,

Miller (2023) noticed softmax attention cannot "zero out" attention heads. More specifically, when $\lim_{x_i \to \inf}$ for all tokens, $x_i$, it is desirable for the head output to produce zero, thus giving no weight to any tokens. In actuality, it produces a uniform distribution over the tokens instead. To fix this issue, Miller (2023) adds a constant to the denominator. This work emphasizes how softmax has subtle problems even though it works well in practice. Smith (2025) asks why attention works, building upon Miller (2023) to change the attention mechanism. However, the focus of this work is on the improvements to attention, rather than exploring exactly what makes attention work. Deng et al. (2023) examines the performance gap between linear and softmax attention, but only in a classification and empirical context. Why linear attention is lacking compared to softmax attention, in general, remains not fully understood. Han et al. (2024) shows that linear attention is lacking an injective property and cannot model local features while softmax attention is injective and can model local features. Collins et al. (2024) uses Lipschitzness to explain in-context learning in softmax. These provide intuition but, ultimately, these properties do not fully explain the expressiveness gap.

Katharopoulos et al. (2020) introduced a formulation for linear attention using recurrent network components. In their work, they also mention that their "*formulation does not impose any constraint on the feature function and it can be used for representing any transformer model, in theory even those using softmax attention.*" This inspired our current work to formulate softmax attention as a RNN and use this formulation to describe what makes softmax attention expressive, in general. Moreover, we attempt to explain exactly why linear attention is not as expressive as softmax attention by showing linear attention approximates a subset of elements that comprise softmax attention.

## 3 Methodology

We first motivate our approach using the traditional formulation of causal softmax attention for calculating the output of the layer, $O_t$, at time $t$:

$$O_t = \text{Softmax}(Q_t \cdot K_{1:t}) \cdot V_{1:t} = \frac{\sum_{s=1}^{t} e^{Q_t \cdot K_s^T} \cdot V_s}{\sum_{n=1}^{t} e^{Q_t \cdot K_n^T}} \tag{1}$$

We would like to rearrange this equation such that one can interpret the attention mechanism in a recurrent form. However, the exponentiation in softmax couples $Q_t$ and $K_s$, preventing a direct regrouping. For instance, in linear attention the attention function is replaced by $\phi(Q_t) \cdot \psi(K_s)^T$. This decoupling allows causal linear attention to be reformulated into a recurrent structure, as follows:

$$O_t = \phi(Q_t) \cdot \sum_{s=1}^{t} \psi(K_s)^T \cdot V_s \quad = \quad \phi(Q_t) \cdot H_t \tag{2}$$

$$\text{where} \quad H_t = \sum_{s=1}^{t} \psi(K_s)^T \cdot V_s \quad = \quad \psi(K_t)^T \cdot V_t + H_{t-1}$$

In order to derive a recurrent representation for softmax, we analyze the numerator and denominator of softmax attention separately. We first examine causal softmax attention without the denominator and show there exists a recurrent form. We use the Taylor series expansion of softmax to achieve this formulation (Section 3.1). Having a recurrent form, Section 3.2 shows linear attention, as in equation 2, is a first order approximation of softmax attention. Appendix B shows another motivating example—the quadratic case—for an intuition of how the recurrent form expands for higher order Taylor series approximations. In Section 3.3, the denominator is reinterpreted, using the language of RNNs with results provided in Section 4.

### 3.1 Recurrent Softmax Attention

Causal softmax attention, equation 1, is composed of exponentials of inner products between the query vector, $Q_t$, and key vectors, $K_s$. Using a decomposed inner product form, causal softmax attention can be rewritten

as an RNN by taking the Taylor series expansion of the exponential function. Here, we set the multiplicative inverse of the denominator to $G_t$ for simplicity. Although the derivation is for the causal case, it can be extended to the bidirectional case as in Appendix C.

Note that, in the explanation below, we make use of several properties:

1. **Decomposed Inner Product Property**: $(A \cdot B)^n$ is equivalent to $(A^{\otimes n}) \cdot (B^{\otimes n})$ by equation equation 7. A full derivation of this property is available in Appendix A.

2. **Inner Product Equivalence**: $\langle A, B \rangle = \sum_{i=1}^{d} (A \odot B)_i = \sum_{i=1}^{d} A_i B_i$

3. **Kronecker Products Shorthand**: $A^{\otimes n} = \bigotimes_{i=1}^{n} A = A \otimes A \otimes \cdots \otimes A \in \mathbb{R}^{d^n}$

where $A, B \in \mathbb{R}^d$, $A \odot B \in \mathbb{R}^d$ is the Hadamard product, $A \cdot B^T \in \mathbb{R}$ is the inner product, $A^T \cdot B \in \mathbb{R}^{d \times d}$ is an outer product, and the lack of an explicit operation denotes either: (1) scalar multiplication with another scalar, vector, or matrix or (2) matrix-vector multiplication, similar to "slicing multiplication" in most numerical packages. With these properties, we can now reformulate softmax attention as follows:

$$
\begin{aligned}
O_t &= G_t \sum_{s=1}^{t} e^{Q_t \cdot K_s^T} V_s \\
&= G_t \sum_{s=1}^{t} \sum_{n=0}^{\infty} \frac{1}{n!} (Q_t \cdot K_s^T)^n V_s && \text{By definition of the Taylor Series of } e \\
&= G_t \sum_{s=1}^{t} \sum_{n=0}^{\infty} \frac{1}{n!} \sum_{i=1}^{d^n} \left(Q_t^{\otimes n}\right)_i \left((K_s^{\otimes n})^T\right)_i V_s && \text{By } equation\ 7 \\
&= G_t \sum_{n=0}^{\infty} \frac{1}{n!} \sum_{i=1}^{d^n} \sum_{s=1}^{t} \left(Q_t^{\otimes n}\right)_i \left((K_s^{\otimes n})^T\right)_i V_s && \text{By rearranging sums} \\
&= G_t \sum_{n=0}^{\infty} \frac{1}{n!} \sum_{i=1}^{d^n} \left(Q_t^{\otimes n}\right)_i \sum_{s=1}^{t} \left((K_s^{\otimes n})^T\right)_i V_s && \text{By factoring out Q} \\
&= G_t \sum_{n=0}^{\infty} \frac{1}{n!} \left(Q_t^{\otimes n}\right) \sum_{s=1}^{t} \left((K_s^{\otimes n})^T\right) \cdot V_s && \text{By inner product equivalence} \\
&= G_t \sum_{n=0}^{\infty} \frac{1}{n!} (Q_t^{\otimes n}) H_t^n, \qquad H_t^n = \sum_{s=1}^{t} ((K_s^{\otimes n})^T) \cdot V_s && \text{Define hidden state}
\end{aligned}
$$

Where $Q \in \mathbb{R}^{N,d}, K \in \mathbb{R}^{M,d}, V \in \mathbb{R}^{M,e}, G \in \mathbb{R}^N, Q_t \in \mathbb{R}^d, K_s \in \mathbb{R}^d, V \in \mathbb{R}^e, G_t \in \mathbb{R}$
$N$ and $M$ are the sequence dimensions, indexed by $t$ and $s$ respectively.
$d$ (and $e$ in Figure 1) are the embedding dimensions where $d$ is indexed by $i$.
$n$ is the $n^{th}$ order term in the Taylor expansion.

Thus, the softmax attention numerator does have a recurrent formulation. This formulation can also be seen visually in Figure 1. Rather than the output coming from a single recurrent equation, softmax is a sum of infinite recurrent outputs, each weighted by $\frac{1}{n!}$. Each of the RNNs that comprise the output of softmax attention have a hidden state of shape $\mathbb{R}^{d^n,e}$ due to the $n^{th}$ order Kronecker product on the queries and keys. The $n^{th}$ order Kronecker product can be thought of as creating $n^{th}$ order multiplicative interactions between dimensions of the keys and queries. The hidden state for each of the infinite RNNs can be thought of as accumulating information of $n^{th}$ order interaction terms on the dimension of $Q$ and $K$.

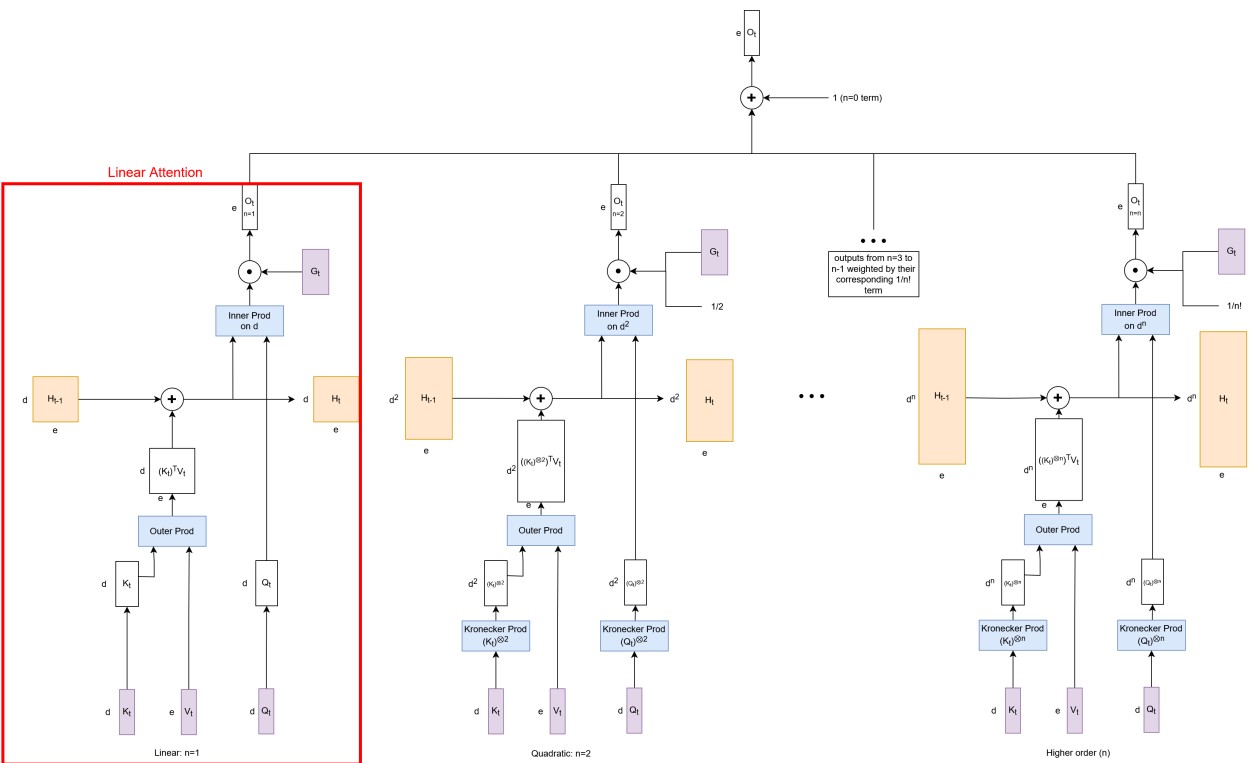

Figure 1: Softmax attention as an RNN. We define $G_t$ in place for the softmax denominator. Linear attention is equivalent to the $n = 1$, first order, term. Expanded plots can be found in Appendix G.

## 3.2 Linear Attention is a First Order Approximation

With this recurrent formulation, notice that taking the $n = 1$ term of the Taylor Series sum results in the well known form of linear attention, as seen in Equation 2.

$$O_t^{(linear)} = \frac{1}{1!}(Q_t^{\otimes 1})\sum_{s=1}^{t}((K_s^{\otimes 1})^T)\cdot V_s = Q_t\sum_{s=1}^{t}K_s^T\cdot V_s = Q_tH_t \qquad H_t = \sum_{s=1}^{t}K_s^T\cdot V_s \qquad (3)$$

Equation 3 and the left portion of Figure 1 show that linear attention is a linear approximation of softmax attention when $\phi = \psi = I_d$. As linear attention is just a single term in softmax attention, this derivation shows how linear attention is a subset of softmax attention and provides an intuitive explanation as to why linear attention is typically less performant. Even when other functions are used for $\phi$ and $\psi$, they can only manipulate a single recurrent chain—they cannot model the higher order terms from softmax. These terms in softmax attention allow it to model combinatorial interactions between inner product dimensions—thus, it is at least as expressive as linear attention. Also notice that each subsequent RNN comprising softmax uses a larger hidden state, modeling higher-order interactions on the combinatorial dimension of $Q$ and $K$. Linear attention works with a single, smaller hidden state operating only on the dimension of $Q$ and $K$, without higher-order interactions. To highlight the difference between softmax attention and linear attention, we also derive a quadratic approximation in Appendix B, which models pairwise interactions on the dimension and has a hidden state quadratically larger than linear attention.

This result raises an important difference between linear attention, with functions on $Q$ and $K$, versus nonlinear attention, which uses a non-decomposable function on the inner product of $Q$ and $K$. Despite the two forms appearing similar, they are fundamentally different in their ability to model higher order interactions. A function on each of the vectors, $\phi(Q_t)$ and $\psi(K_s)$ restricts the vector space dimensions of $Q_t$ and $K_s$. For example, *ReLU* restricts vectors to the positive portion of the vector space. On the other hand,

the exponential function on the inner product space does not restrict the vector space. Instead, this function creates $n^{th}$ degree multiplicative factors between dimensions of $Q_t$ and $K_s$ for all $n \in [0, \infty)$. Practically, these multiplicative dimensions become less influential for larger $n$, as they are weighted by $\frac{1}{n!}$, but they cannot be disregarded entirely.

### 3.3 Reinterpreting the Denominator

In traditional softmax attention the denominator is calculated as:

$$G_t = \frac{1}{\sum_{s=1}^{t} e^{Q_t \cdot K_s^T}} \tag{4}$$

where $G_t$ can have values from 0 to 1 [2]. Typically, these values are calculated from the same $Q_t$ and $K_s$ values as the numerator. However, we hypothesize that the exact normalization is not the most crucial aspect of $G_t$. Rather, we observe that $G_t$ could be interpreted like a gate or norm that stabilizes the numerator, especially for long contexts when $t$ becomes large. We note that this re-interpretation of the denominator may not be strictly accurate, but does provide a convenient mechanism for capturing some aspects of the softmax denominator. This stabilization aspect is hypothesized to be the crucial function of $G_t$, rather than the exact form of $G_t$. Therefore, we reformulate $G_t$ as a gate via:

$$(Gate) \qquad \frac{1}{t} G_t \sum_{s=1}^{t} e^{Q_t \cdot K_s^T} V_s = \sum_{n=0}^{\infty} \frac{1}{n!} (Q_t^{\otimes n}) H_t^n, \qquad H_t^n = \frac{1}{t} G_t \sum_{s=1}^{t} ((K_s^{\otimes n})^T) \cdot V_s \tag{5}$$

This assumption of the role of $G_t$ allows us to approximate it like an output gate in a recurrent structure at time $t$. This representation also ties the softmax attention mechanisms to traditional properties of expressive recurrent networks, such as the LSTM Hochreiter and Schmidhuber (1997) and GRU Cho et al. (2014). However, this interpretation as a gate may be too broad of an assumption as it also allows the sequence to grow without bound. This issue can be mitigated by dividing by the sequence length, clamping the inner product (for example, we clamp the pre-exponential value to 5), and clipping the gradient. While these tricks help ensure the gate is numerically stable, they are not particularly elegant solutions and complicate the overall implementation. Alternatively, $G_t$ can be interpreted as a norm at time $t$, normalizing the numerator according to its length and values. This can be realized via:

$$(Norm) \qquad \left\| \sum_{s=1}^{t} e^{Q_t \cdot K_s^T} V_s \right\| = \left\| \sum_{n=0}^{\infty} \frac{1}{n!} (Q_t^{\otimes n}) H_t^n \right\|, \qquad H_t^n = \sum_{s=1}^{t} ((K_s^{\otimes n})^T) \cdot V_s \tag{6}$$

where $\|\cdot\|$ denotes a vector norm. The optimal type of norm is not immediately clear, and could be any number of formulations such as $L_2$, $RMS$, or others. We test each interpretation of $G_t$ under a recurrent perspective, as either a gate or a norm, to understand its equivalence to the traditional softmax denominator.

## 4 Results

To evaluate our hypothetical softmax alternative, we train multiple Llama 2 Touvron et al. (2023) Touvron et al. (2023) models for next token language modeling. Keeping the rest of the architecture constant, we replace the attention mechanism with the proposed variations. We show log loss to emphasize the differences between each model. Section 4.1 shows our proposed replacement is empirically equivalent to softmax. Section 4.2 examines the scalability of the proposed replacement. Section 4.3 evaluates linear attention against normal softmax and our proposed methods. As our method uses a Taylor expansion, Section 4.4 looks at the performance of different linear attention variations via progressive additions of higher order powers. Section 4.5 ablates various elements of the recurrent softmax attention.

---

[2] $G_t$ can be greater than 1 if the denominator is less than 1, which is rare, only occurring for small $t$.

### 4.1 Softmax Equivalence

**Experimental Setup**  In our experiments, we test both replacing the denominator with a gate as in equation 5 and with a norm as in equation 6 alongside normal softmax attention. To evaluate the applicability to various domains, we retrain on three datasets: The Pile Gao et al. (2021), SlimPajama Shen et al. (2023), and FineWeb Penedo et al. (2024). The Pile is a dataset created by Eleuther AI composed of 825 GiB of English text on various domains such as code, technical papers, math, and articles. SlimPajama is a 627 billion token dataset that is a cleaner subset of RedPajama Weber et al. (2024), a dataset of various web crawl data. FineWeb is a cleaned and de-duplicated 5-trillion token dataset compiled from 96 different common crawl snapshots. Each tested model is about 300 million parameters and trained on a sequence length of 1024. Adding a gate or norm requires up to an additional $d$ parameters per layer, which is insignificant relative to the model size. More hyperparameters for our models can be found in Appendix D.

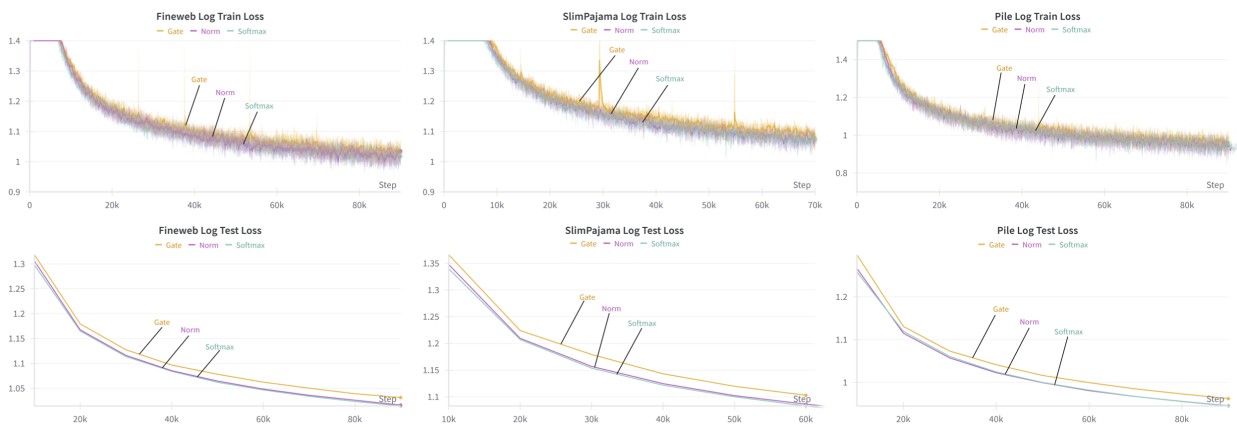

Figure 2: Test and train loss on various datasets for softmax attention and the proposed methods with gate or norm replacements. Expanded plots can be found in Appendix G.

**Result**  Figure 2 shows the resulting loss curves for each dataset and model variant. The loss for the normed model variant follows the model trained with native softmax attention precisely while the model with a gate performs slightly worse. We employ the $L_2$ vector norm for this analysis and find that it is numerically stable. The gate was semi-unstable during training. We noticed recoverable loss spikes during training where the model loss would spike and resume after some number of steps. Dividing by the sequence length, clipping the inner product before exponentiation, and performing gradient clipping, helped produce fewer spikes, however some instability was still observed. This result implies that the function of $G_t$ is most similar to a norm operation. Moreover, the norm does not need to mirror the traditional softmax exponentiation—a simple vector norm approximates this well. We note, however, that this does not prove the denominator can be a simple recurrent gate or norm—we only claim that performance is similar.

### 4.2 Scaling

To investigate if scaling laws hold, we scale the model in two ways. We scale the model from 300M parameters to 2B parameters, keeping all other hyperparameters constant to evaluate the scalability of the model itself. We also scale the sequence length from 1024 to 4096, keeping all other hyperparameters constant to examine how our method scales with longer sequences. These scaling are tested upon the FineWeb Penedo et al. (2024) dataset. The scaled comparisons are found in Figure 3, showing our proposed attention formulation scales identically to softmax (with both the sequence length and the model size). Again we observe that the norm better mirrors softmax performance and is more numerically stable.

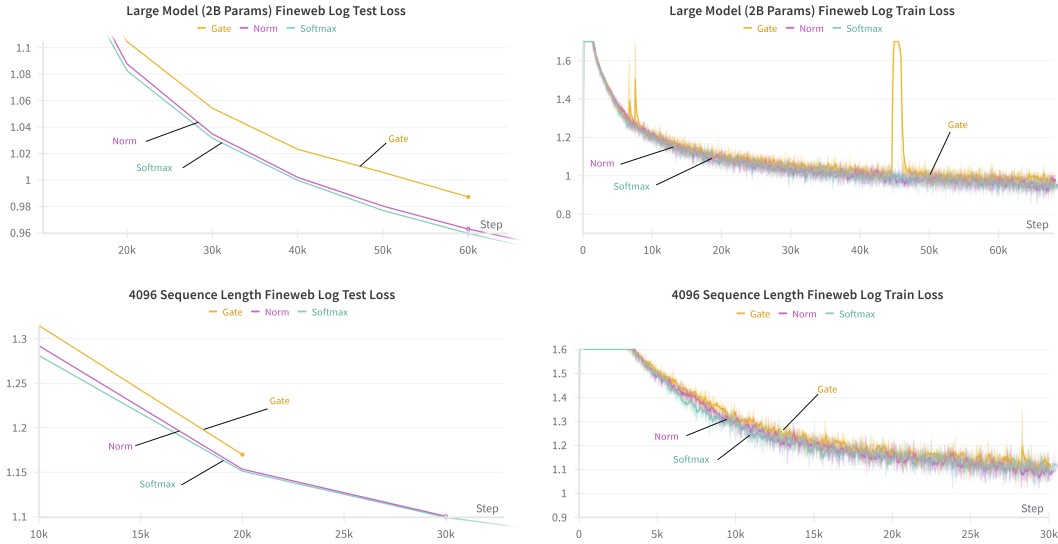

Figure 3: Test and train loss on the FineWeb dataset for large models (about 2B parameters) on 1024 sequence length and small models (about 300 million parameters) on 4096 sequence length for softmax attention and the proposed methods with gate or norm replacements. Some experiments were cut short due to time constraints

### 4.3 Linear Attention

As the proposed softmax decomposition has a direct relationship to linear attention, we evaluate the method against several variations of linear attention. Figure 4 shows that softmax attention and our proposed methods outperform each linear attention variant by a significant gap. Additional details are provided in Appendix F. This behavior supports the assertions in Section 3.2, showing linear attention is a subset of softmax attention.

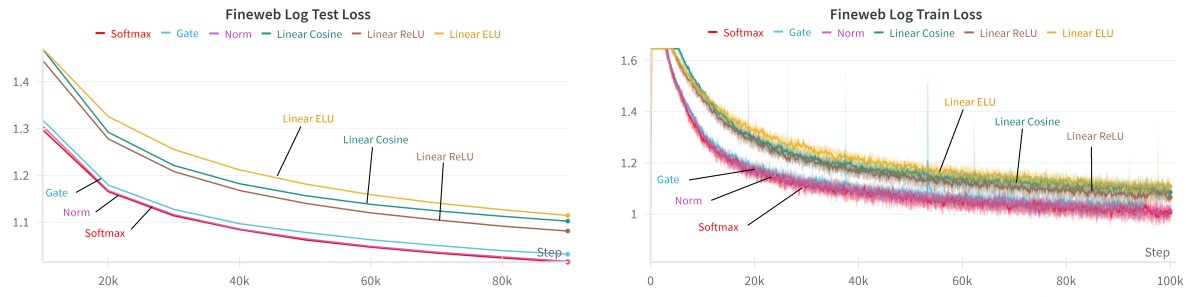

Figure 4: Test and train loss on the FineWeb dataset for various linear attention methods, softmax attention, and the proposed methods with gate or norm replacements.

### 4.4 Taylor Series Terms

As our method uses a Taylor series, we investigate how performance behaves as higher order terms are added to the less expressive attention variations. That is, we use the recurrent formulation for a particular type of linear attention and gradually add in higher order terms from the softmax Taylor expansion of softmax. The results are shown in Figure 5 for three types of linear attention: cosine similarity Mongaras et al. (2025), ReLU Xie et al. (2025), and $elu(x) + 1$ kernel Katharopoulos et al. (2020). As more terms are added to the softmax approximation, the performance smoothly transitions from "linear attention performance"

to "softmax attention performance." After adding in terms up to $n = 10$, we observe that the recurrent approximation mirrors softmax with negligible differences. For the linear attention variants, however, we find that adding more terms (described in Appendix F) improves performance but never fully reaches the performance of softmax attention (even for $n = 10$). We hypothesize this performance gap exists because linear attention variants have functions on the independent vectors, $\phi(Q)$ and $\psi(K)$, restricting the reachable vector space when combined; whereas softmax does not restrict this vector space. We leave exploring this observation to future work. We note that cosine attention does not gain any benefit from additional terms and hypothesize that due to inner product values being between 0 and 1, the resulting higher order interaction terms are less than 1, limiting the magnitude of higher order terms and therefore the impact of these higher order terms on the resulting output.

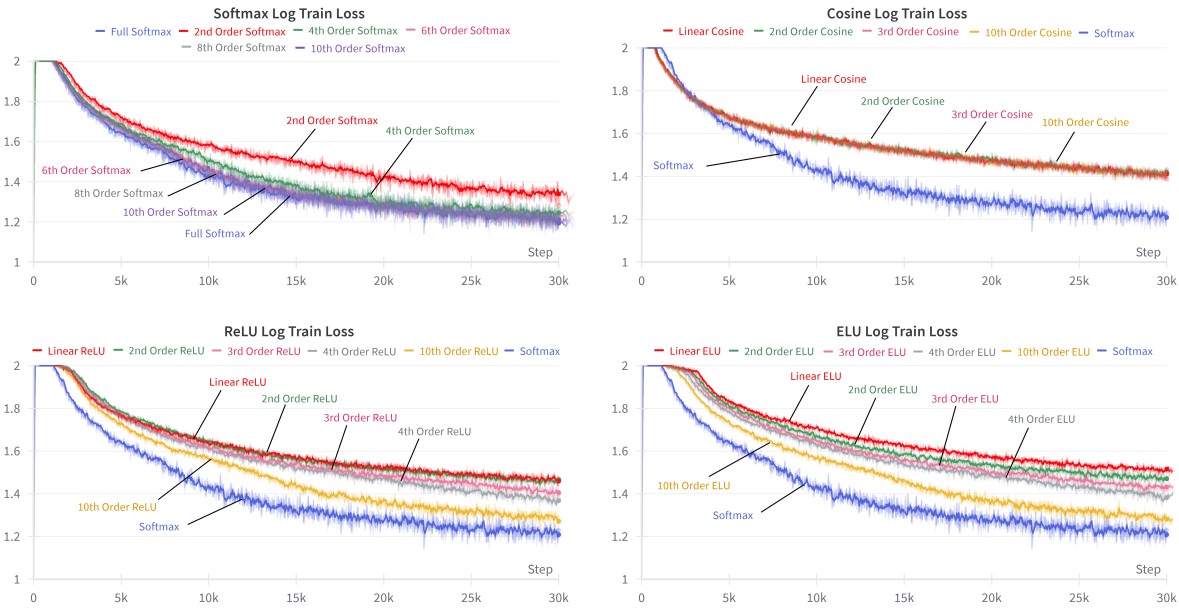

Figure 5: Log train loss for softmax attention and various linear attention method when summing more powers of the inner product. The nth order denotes the sum of powers from 0 to n.

## 4.5 Ablation Analysis

To further investigate various recurrent elements, we conduct an ablation study using the 300M parameter model, varying both the gated and normed variants of our method. The results of these ablations are plotted in Figure 6. The leftmost plots ablate several elements of the recurrent softmax denominator, $G_t$, by removing, detaching, and dividing by the sequence length, $S$ (with varying combinations of all three). We find that detaching the denominator from the computation graph significantly hurts downstream performance. Removing the denominator and dividing by the sequence length gives similar performance to softmax, but also creates instability. Adding a gate helps to somewhat stabilize the training, though not entirely. Finally, we find that removing the denominator and adding a norm is what appears to mirror softmax most closely.

**Gate**  The middle column of plots in Figure 6 ablates the gate method employed by combining sequence length normalization with an input gate or output gate (more fully explained in Appendix E), and investigating ReLU linear attention with a gate. The results shows that having a gate helps to stabilize training loss, but must be accompanied by sequence length normalization for good test performance. Furthermore, replacing the exponential with a decomposable ReLU kernel significantly hurts performance, regardless of gating or sequence length normalization. This result emphasizes the importance of the exponential function on the inner product of $Q$ and $K$, as well as indicating that some sort of sequence length normalization is required.

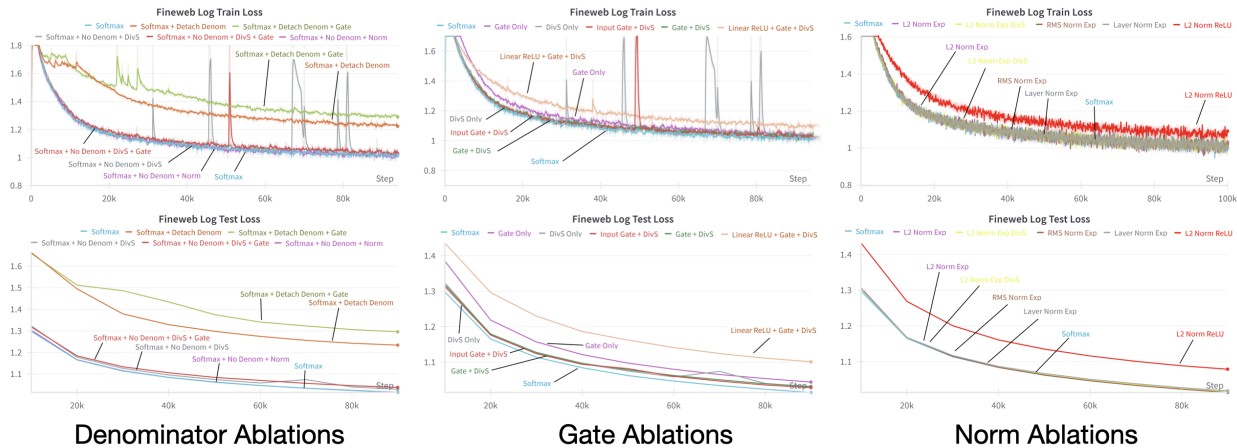

Figure 6: Test and train loss on various datasets for softmax attention and the proposed methods with gate or norm replacements. Expanded plots can be found in Appendix G.

**Norm** The rightmost column in Figure 6 varies the norm method employed, investigating the $L_2$ norm, RMS norm, layer norm, and $L_2$ norm with sequence normalization. We also investigate the use of an $L_2$ norm with the decomposable ReLU kernel. The results shows that any type of normalization, with or without learnable parameters, works just as well as softmax. However, replacing the exponential with a decomposable ReLU kernel significantly degrades performance. This graph further emphasizes that the important aspects of softmax attention are an exponential plus a vector norm. The choice of norm does not appear to influence training stability or test performance. However, there are differences between a vector norm (e.g., $L_2$) and a sequence length normalization (e.g., division by $S$ or gating). The vector norm appears to achieve the best performance, implying it is a necessary aspect of softmax, but the exact form seems to be unimportant.

## 5 Conclusions and Limitations

This work connects linear and softmax attention under a unifying recurrent formulation. A Taylor series expansion of the softmax attention numerator was employed to develop a recurrent form and competing hypotheses for approximating the softmax denominator were investigated. Linear attention was shown to be a first order approximation of softmax. Using the recurrent softmax attention formulation, equivalence was shown empirically and the crucial elements of softmax attention were analyzed in an ablation study. Additional experiments showed that a tenth order Taylor series approximation is sufficient similar to softmax. Finally, different types of normalization and gates were explored showing that any vector norm is sufficient to approximate the functionality of the softmax attention denominator. The theory developed in this work is crucial to understanding the performance bounds of softmax compared to other forms of attention. Moreover, this theory may be employed to uncover more performant or efficient attention mechanisms.

**Limitations** The current formulation covers only linear attention and softmax attention, future work can expand it to more complicated recurrent architectures such as RWKV Peng et al. (2025) and state-space models like Mamba Dao and Gu (2024). As recurrent architectures similar to RWKV and Mamba are extensions of linear attention, the theory developed in this work should extend to the additions made in these works. Only the causal next token prediction task was investigated. While the derivations should generalize to other domains, both bidirectional and causal, further investigation is necessary to ensure this generalization. Future work can also incorporate changes from these recurrent models to softmax attention to potentially improve efficiency or performance of softmax attention implementations. Finally, the hyperparameters for the architectures presented were not exhaustively investigated, which could influence the final loss values.

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

# A    Inner Product Decomposition

Let $A, B \in \mathbb{R}^d$

Define $A \odot B \in \mathbb{R}^d$ as the Hadamard product of A and B.

Define $A \cdot B \in \mathbb{R}$ as the inner/dot product of A and B

Note that the inner product is defined as $A \cdot B = \sum_{i=1}^{d} (A \odot B)_i = \sum_{i=1}^{d} A_i B_i$

Define $A^{\otimes n} = \bigotimes_{i=1}^{n} A = A \otimes A \otimes \cdots \otimes A \in \mathbb{R}^{d^n}$ as n - 1 Kronecker products

The lack of an explicit operation denotes either: (1) scalar multiplication with another scalar, vector, or matrix. or (2) matrix-vector multiplication, similar to "slicing multiplication" in most numerical packages.

Using the notation mention in Section 3.1, we show the decomposed form for an inner product space to the $n^{th}$ power is equivalent to the inner product of $n^{th}$ order Kronecker products of each vector.

$$(A \cdot B)^n$$

$$= \left[ \sum_{i=1}^{d} (A \odot B)_i \right]^n \qquad \text{By equation } equation \ 2$$

$$= \left[ \sum_{i=1}^{d} (A \odot B)_i \right] \left[ \sum_{i=1}^{d} (A \odot B)_i \right] \cdots \left[ \sum_{i=1}^{d} (A \odot B)_i \right] \qquad \text{Power to product of n terms}$$

$$= \sum_{i=1}^{d} \sum_{j=1}^{d} \cdots \sum_{k=1}^{d} \underbrace{(A \odot B)_i (A \odot B)_j \ \cdots \ (A \odot B)_k}_{\text{combinatorial multiplication from } i \ \cdots \ k} \qquad \text{By rearranging the sums}$$

$$= \sum_{i=1}^{d^n} \left[ (A \odot B)^{\otimes n} \right]_i \qquad \text{By equation } equation \ 3$$

$$= \sum_{i=1}^{d^n} \left[ \left( A^{\otimes n} \right) \odot \left( B^{\otimes n} \right) \right]_i \qquad \text{Rearrange the nth order Kronker Product} \qquad (7)$$

$$= \left( A^{\otimes n} \right) \cdot \left( B^{\otimes n} \right) \qquad \text{Change to inner product by equation } equation \ 2 \qquad (8)$$

## B    Quadratic Derivation

To provide a simple example of the recurrent form derived in Section 3.1, we note that $A^{\otimes 2} = A \otimes A \in \mathbb{R}^{d^2}$ and show the recurrent form.

$$
\begin{aligned}
O_t &= \sum_{s=1}^{t} (Q_t \cdot K_s^T)^2 V_s \\
&= \sum_{s=1}^{t} \left[ \sum_{i=1}^{d} (Q_t \odot K_s)_i \right]^2 V_s && \text{By equation } equation\ 2 \\
&= \sum_{s=1}^{t} \sum_{i=1}^{d^2} (Q_t \otimes Q_t)_i \, (K_s \otimes K_s)_i \, V_s && \text{By equation } equation\ 7 \\
&= \sum_{i=1}^{d^2} \sum_{s=1}^{t} (Q_t \otimes Q_t)_i \, (K_s \otimes K_s)_i \, V_s && \text{rearrange summations} \\
&= \sum_{i=1}^{d^2} (Q_t \otimes Q_t)_i \sum_{s=1}^{t} (K_s \otimes K_s)_i \, V_s && \text{By factoring out Q} \\
&= (Q_t \otimes Q_t) \sum_{s=1}^{t} \left( (K_s^T) \otimes (K_s^T) \right) \cdot V_s && \text{By equation equation 2} \\
&= (Q_t \otimes Q_t) H_t \qquad H_t = \sum_{s=1}^{t} \left( (K_s^T) \otimes (K_s^T) \right) \cdot V_s \in \mathbb{R}^{d^2, d} && \text{Define hidden state}
\end{aligned}
$$

## C    Bidirectional Derivation

$$
\begin{aligned}
O_t &= G_t \sum_{s=1}^{N} e^{Q_t \cdot K_s^T} V_s \\
&= G_t \sum_{s=1}^{N} \sum_{n=0}^{\infty} \frac{1}{n!} (Q_t \cdot K_s^T)^n V_s && \text{By definition of the Taylor Series of } e \\
&= G_t \sum_{s=1}^{N} \sum_{n=0}^{\infty} \frac{1}{n!} \sum_{i=1}^{d^n} \left( Q_t^{\otimes n} \right)_i \left( (K_s^{\otimes n})^T \right)_i V_s && \text{By equation } equation\ 7 \\
&= G_t \sum_{n=0}^{\infty} \frac{1}{n!} \sum_{i=1}^{d^n} \sum_{s=1}^{N} \left( Q_t^{\otimes n} \right)_i \left( (K_s^{\otimes n})^T \right)_i V_s && \text{By rearranging sums} \\
&= G_t \sum_{n=0}^{\infty} \frac{1}{n!} \sum_{i=1}^{d^n} \left( Q_t^{\otimes n} \right)_i \sum_{s=1}^{N} \left( (K_s^{\otimes n})^T \right)_i V_s && \text{By factoring out Q} \\
&= G_t \sum_{n=0}^{\infty} \frac{1}{n!} \left( Q_t^{\otimes n} \right) \sum_{s=1}^{N} \left( (K_s^{\otimes n})^T \right) \cdot V_s && \text{By equation equation 2} \\
&= G_t \sum_{n=0}^{\infty} \frac{1}{n!} (Q_t^{\otimes n}) H_t^n, \qquad H_t^n = \sum_{s=1}^{N} ((K_s^{\otimes n})^T) \cdot V_s \in \mathbb{R}^{d^n, e} && \text{Define hidden state}
\end{aligned}
$$

The only difference between the causal and bidirectional formulations is the range on the sequence sum indexed by $s$. In the causal case, the index ends at $t$ while the bidirectional case sums to the end of the sequence. This means the hidden state operates on the entire sequence for each token rather than just the past $s \leq t$.

## D  Model Parameters

Unless otherwise mentioned, the below are the parameters we used in our models. As our base model is llama 2 Touvron et al. (2023). RoPE Su et al. (2024) is used on the attention matrix and the MLPs follow SwiGLU Shazeer (2020).

1. batch size - 36

2. learning rate - 1e-4

3. warmup steps - 10,000

4. warmup type - linear warmup from 0, linear decay

5. num steps - 100,000

6. precision - float32 and bfloat16 mixed precision

7. Weight decay - 0.01

8. Max sequence length - 1024 for general experiments, 4096 for length scaling experiment

9. Test percentage - 0.001

10. Optimizer - AdamW

11. Adam betas - 0.9 and 0.999

12. Hidden size - 1024 (3072 for the large model)

13. MLP intermediate size - 2048 (6144 for the large model)

14. Num attention heads - 16

15. Num hidden layers - 20

16. Tokenizer - llama2-7b-hf

17. Gradient clipping - 1.0 clipping for gated models, no clipping for all other experiments

Each model was trained for a maximum of 2 days. For most experiments, we use distributed data parallel processing to train on two 80 GB, A100 GPUs with the exception of the large model, trained on 4 GPUs, and 4096 sequence length, trained on 6 GPUs.

## E  Additional Gate Information

The input and output gates referenced in this paper are very similar to that of an LSTM Hochreiter and Schmidhuber (1997) and similar to that used by Qiu et al. (2025). The input gate controls how much information is being added to the LSTM hidden state while the output gate modulates the output of the LSTM cell. As seen in Figure 7, these gating mechanisms can be translated to the original linear attention formulation where the input gate is a scalar between [0, 1] at time $t$, modulating the $K^T V$ outer product while the output gate is a scalar between [0, 1] at time $t$ modulating output after the $Q_t H_t$ inner product. These ideas extend to softmax attention, where it can be computed as an infinite sum of these RNNs—although the

implementation is intractable. However, the gates can be translated to multiplicative values on the $QK^T$ attention matrix. As the rows/queries define the output at time $t$, the output gate can be defined along the rows. Similarly, as the columns define the input at time $s$, the input gate can be defined along the columns. This idea is equivalent to applying the output gate after performing the full attention operation and applying the input gate to the values. Mathematically, this can be expressed as follows:

$$O_t = \sum_{s=0}^{t} \left[ G_t^{in} e^{Q_t \cdot K_s^T} G_s^{out} \right] \cdot V_s \qquad = G_t^{in} \sum_{s=0}^{t} e^{Q_t \cdot K_s^T} \left[ G_s^{out} V_s \right]$$

$$O = \left[ G^{in} \odot e^{Q \cdot K^T} \odot M \odot G^{out} \right] \cdot V \qquad = G^{in} \odot \left[ e^{Q \cdot K^T} \odot M \right] \left[ G^{out} \odot V \right]$$

Where $Q \in \mathbb{R}^{N,d}, K \in \mathbb{R}^{M,d}, V \in \mathbb{R}^{M,e}, G^{in} \in [0,1]^N, G^{out} \in [0,1]^M$

And $Q_t \in \mathbb{R}^d, K_s \in \mathbb{R}^d, V_s \in \mathbb{R}^e, G_t^{in} \in [0,1], G_s^{out} \in [0,1]$

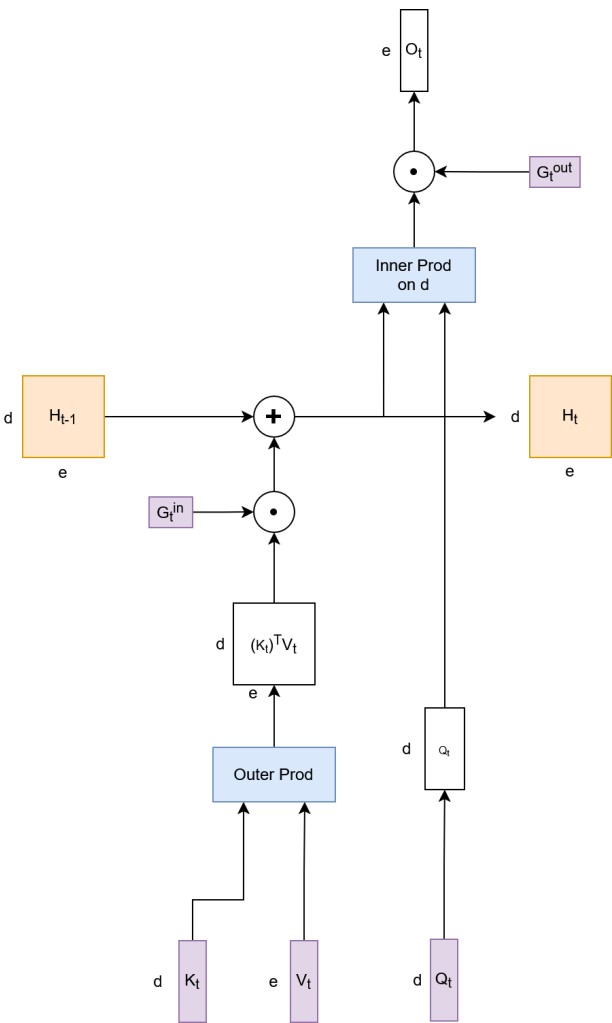

Figure 7: Linear attention as an RNN with an input and output gate.

## F   Linear Attention Expansions

This section provides details on how linear methods are augmented from their base implementation, as presented in Section 4.3. For linear attention variants the activation functions, $\phi$ and $\psi$, are applied to $Q$ and $K$ as in the native linear attention.

$$(Softmax) \qquad O_t = \sum_{n=0}^{\infty} \frac{1}{n!} \left( Q_t^{\otimes n} \right) \cdot \sum_{s=1}^{t} \left( (K_s^{\otimes n})^T \right) V_s \qquad\qquad = \sum_{s=1}^{t} \sum_{n=0}^{\infty} \frac{1}{n!} (Q_t \cdot K_s^T)^n V_s$$

$$(Linear) \qquad O_t = \sum_{n=0}^{\infty} \frac{1}{n!} \left( \phi(Q)_t^{\otimes n} \right) \cdot \sum_{s=1}^{t} \left( (\psi(K)_s^{\otimes n})^T \right) V_s \qquad\qquad = \sum_{s=1}^{t} \sum_{n=0}^{\infty} \frac{1}{n!} (\phi(Q)_t \cdot \psi(K)_s^T)^n V_s$$

# G    Expanded Figures

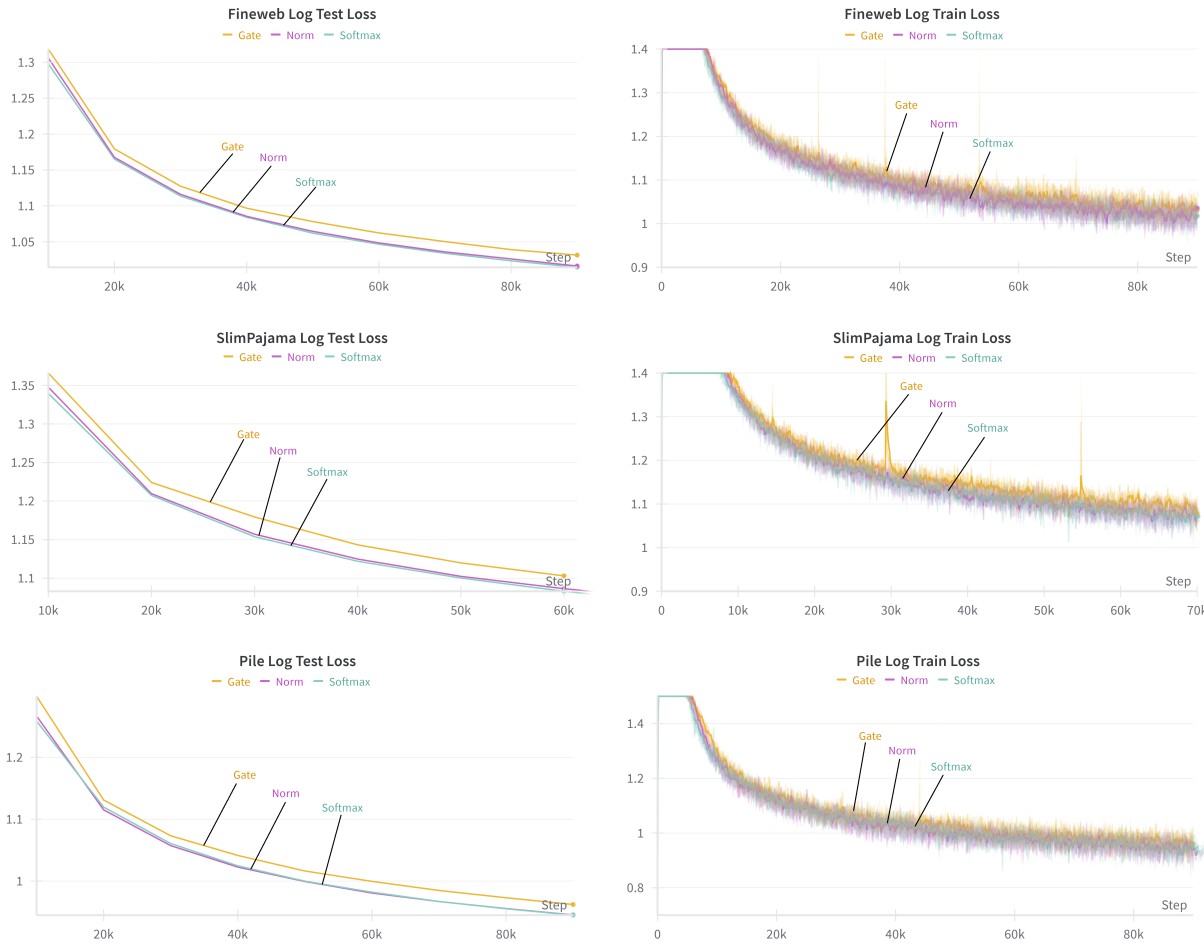

Figure 8: Expanded test and train loss plots on various datasets for softmax attention and the proposed methods with gate or norm replacements. (expanded Figure 2)

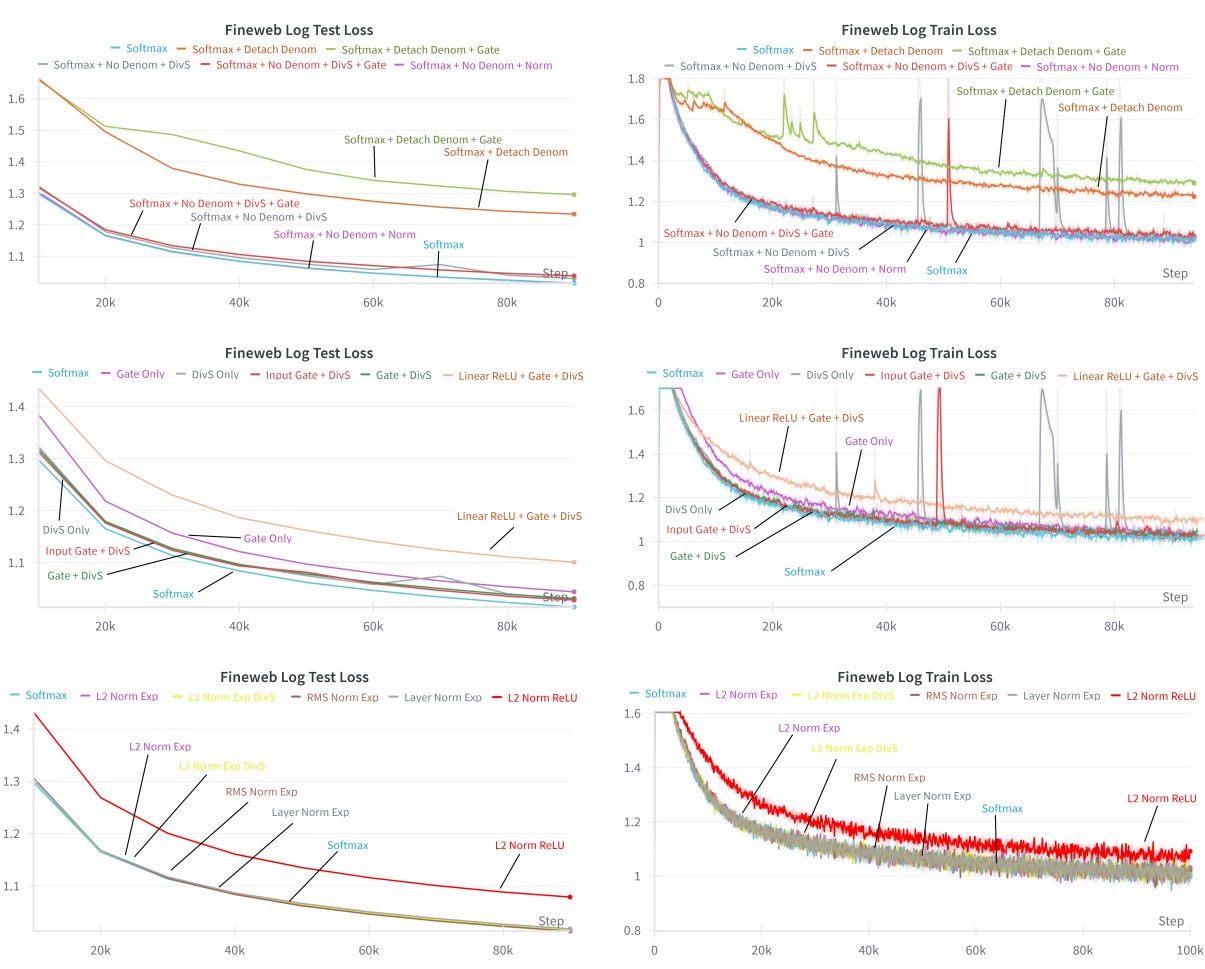

Figure 9: Expanded test and train loss on various datasets for softmax attention and the proposed methods with gate or norm replacements. (expanded Figure 6)

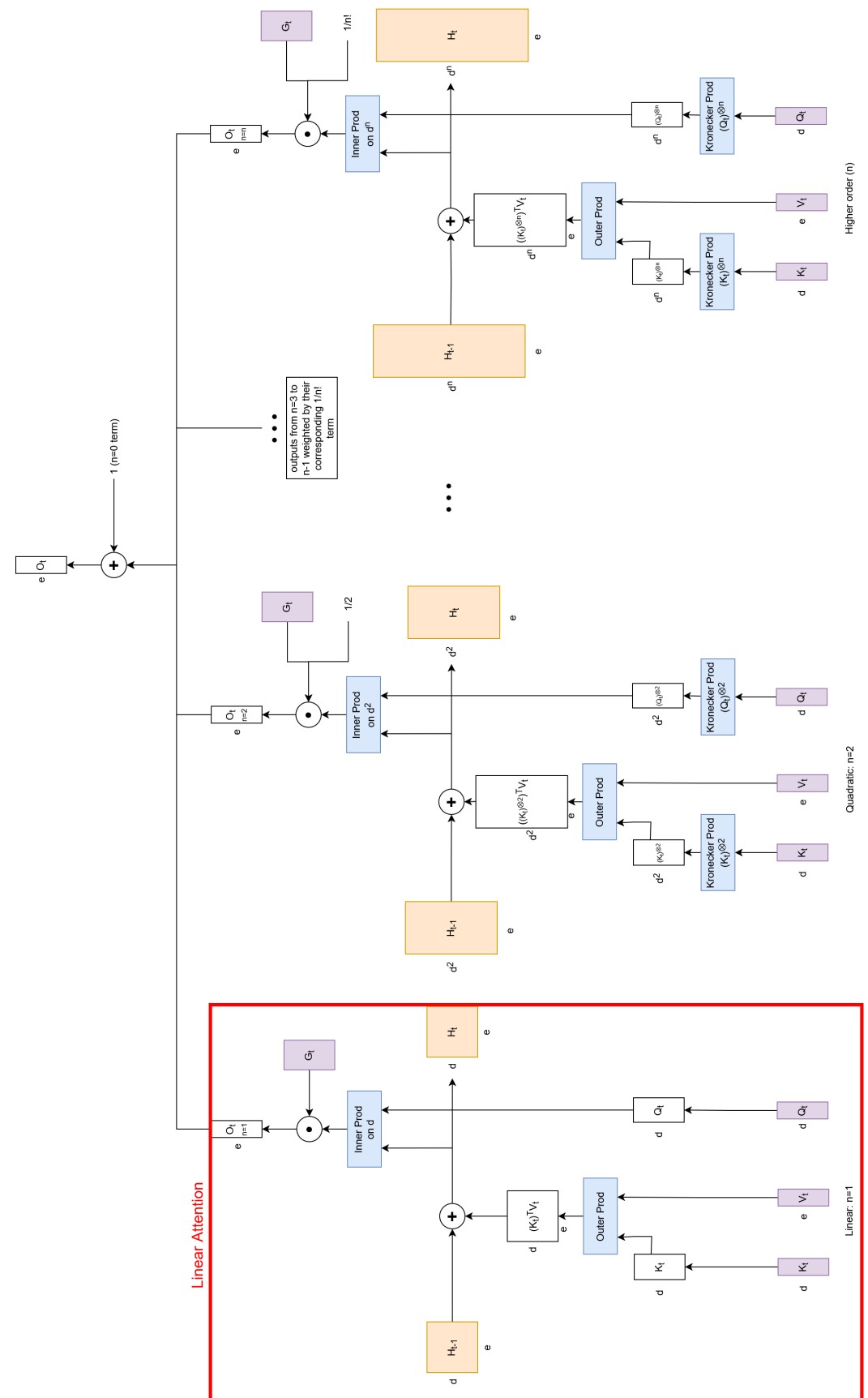

Figure 10: Expanded version of Figure 1. Softmax attention as an RNN. We define $G_t$ in place for the softmax denominator. Linear attention is equivalent to the $n = 1$, first order, term.

