# OpenReview forum: "On the Expressiveness of Softmax Attention: A Recurrent Neural Network Perspective"
_TMLR — Accepted by TMLR_

### Review · Reviewer_PB6Z · 2025-08-17

**Summary Of Contributions:**

This paper provides both theoretical and empirical insights into why softmax attention is more expressive than linear or kernelized variants. By deriving a recurrent formulation via Taylor expansion, the authors show that linear attention corresponds to the first-order approximation, while softmax can be interpreted as a sum of infinite recurrent chains that capture higher-order query–key interactions. They further analyze the role of the denominator, proposing gating and normalization perspectives, and validate these hypotheses through experiments across multiple datasets and model scales. The results suggest that softmax’s superiority arises from its capacity to model higher-order interactions and the stabilizing effect of normalization. In addition, the authors train several LLaMA-2 variants (300M to 2B) to support their empirical findings.

**Audience:**

Yes

**Audience Explanation:**

The paper provides detailed theoretical proofs and comprehensive experiments to substantiate its claims

**Claims And Evidence:**

Yes

**Claims Explanation:**

Strength：
1. The paper addresses a meaningful and timely question.
2. It provides both theoretical analysis and empirical validation of the differences between softmax and linear attention, with solid and comprehensive content.

Weakness
1. Although the experiments align learning rates across LLaMA-2 variants for fair comparison, prior work suggests that the optimal learning rate for linear attention may differ from that of softmax. Under this setting, can the reported results truly reflect the best achievable performance of linear attention?

**Requested Changes:**

1. Could you discuss the impact of hybrid parameters on the experimental validation?

---

> ### Author Response · Authors · 2025-09-19
> **Author Response**
>
> Thank you for bringing up these points. When initially testing between linear attention and softmax attention, we did vary the learning rate for both across runs. In our initial tests, the learning rate didn't drastically influence training. Therefore, we did not include this ablation in the final analysis. However, in our current results we do include a linear learning rate scheduler (warmup for 1K steps and decays across timesteps).
>
> While we do not include hyperparameter experiments, we do include various ablations across various gating styles, normalization, linear attention activation functions, and different model orders. Other hyperparameters such as the learning rate scheduler and Adam beta values were kept constant. While we do have moderate resources for computations, it may not be possible for us to provide an extensive search across several hyperparameters as this analysis would likely take a few months to complete. To partially address this comment, we have added lack of hyperparameter investigation as a limitation of the work.

---

### Review · Reviewer_7CMB · 2025-08-18

**Summary Of Contributions:**

The paper presents a novel theoretical analysis of the softmax attention mechanism from the perspective of recurrent neural networks. The primary contribution is the derivation of a recurrent formulation for the numerator of the softmax attention calculation. This is achieved by applying a Taylor series expansion to the exponential function, which reframes the attention mechanism as an infinite weighted sum of RNN-like components. It provides a principled explanation for the performance gap between softmax and linear attention, showing that linear attention is equivalent to only the first-order term of this infinite series, thus lacking the capacity to model higher-order feature interactions. This paper also deconstructs the expressiveness of softmax attention into an ensemble of recurrent structures, each capturing progressively higher-order interactions between queries and keys. The authors also recognize that the softmax denominator does not lend itself to a similar recurrent derivation. They hypothesize that its primary function is normalization and stabilization, rather than its exact mathematical form being crucial. They test this by replacing the denominator with simpler mechanisms, most notably a standard vector norm.

Pros: The theoretical connection established between softmax and an infinite ensemble of RNNs is novel, providing a clear and unifying perspective for understanding one of deep learning's most important mechanisms. The work offers clear explanations for why softmax attention is more expressive than its linear counterparts. Extensive experiments on multiple large-scale datasets and model sizes support the claims. The finding that a simple L2 norm can replace the softmax denominator with virtually no performance degradation is a significant and noteworthy result.

Cons: The paper's primary weakness is that it does not derive a complete recurrent equivalent for the entire softmax mechanism. The treatment of the denominator is a functional approximation, not a mathematical derivation. While the authors are transparent about this in the methodology section, the title and abstract could be interpreted as claiming a full, exact equivalence, which might be misleading.

**Additional Comments:**

NA.

**Audience:**

Yes

**Audience Explanation:**

This research addresses a high-impact question at the heart of modern deep learning: "What makes softmax attention so effective?" The work provides a new theoretical foundation for understanding the expressiveness of the core component of Transformer models, making it highly relevant to Transformer and Attention Researchers. Moreover, the paper builds an explicit bridge between the Transformer and RNN worlds. Researchers working on state-space models and recurrent architectures like RWKV, Mamba, and other SSMs will likely be interested in this unifying perspective.

**Broader Impact Concerns:**

NA.

**Claims And Evidence:**

Yes

**Claims Explanation:**

The paper's claims are well-supported, with evidence of two distinct types that the authors handle appropriately.
1. The central claim that the softmax numerator can be expressed as an infinite sum of recurrent components is supported by a rigorous and sound mathematical derivation in Section 3.1 and Appendix A. The steps, involving a Taylor series expansion and the application of an identity involving Kronecker products, are mathematically correct.
2. The second major claim—that the denominator's function is primarily normalization and can be replaced by a vector norm—is supported by extensive and convincing empirical evidence. Figure 2 shows that the "Norm" variant's loss curves are nearly indistinguishable from the standard Softmax baseline across three different large-scale datasets. Furthermore, Figure 3 demonstrates that this equivalence holds when scaling up both the model size and the sequence length. Finally, the ablation analysis in Section 4.5 and Figure 6 further strengthens this claim by showing that various types of vector norms perform well, while removing normalization entirely or using improper gating leads to instability or performance degradation.

**Requested Changes:**

I suggest the authors slightly revise the language in the Abstract and Introduction to more precisely reflect the distinction between the exact derivation for the numerator and the functional approximation for the denominator. For example, instead of "This work demonstrates that linear attention is an approximation of softmax attention by deriving the recurrent form of softmax attention," a more precise phrasing might be: "This work demonstrates that linear attention is a first-order approximation of the softmax numerator by deriving its full recurrent form. We further show empirically that the denominator's function can be effectively replaced by a simple vector norm."

In Section 3.1, the dot product symbol is used to represent several different operations. While context makes it understandable, explicitly defining the operations or using different symbols in the final recurrent formula would improve mathematical clarity.

Some labels and lines in the figures are small and can be difficult to distinguish.

---

> ### Author Response · Authors · 2025-09-19
> **Author Response**
>
> Thank you for reviewing our submission and for these suggestions. We have made the suggested changes regarding the denominator and ambiguity of the dot product operator. We have additionally increased the size of some of the figures to fill the page and have included larger variations of some of the smaller images in the appendix as to not exceed space limitations in the main content.

---

### Review · Reviewer_LbKq · 2025-09-06

**Summary Of Contributions:**

The paper derives a recurrent reformulation of softmax attention via a Taylor-series expansion of the exponential in the QK inner product, yielding an interpretation as an infinite sum of RNNs with growing-order hidden states. From this lens, it argues (and demonstrates empirically) that (i) standard “linear attention” corresponds to the first-order term and is therefore less expressive, and (ii) the softmax denominator primarily serves as a stabilizing factor that can be replaced by a simple vector norm (with a gate being possible but less stable). Experiments on next-token LM with LLaMA-like models (~300M and ~2B) trained on The Pile, SlimPajama, and FineWeb suggest the proposed “normed” formulation matches softmax training curves, while linear-attention variants lag; adding higher-order terms (up to n=10) smoothly bridges the gap toward softmax.

**Audience:**

Yes

**Audience Explanation:**

Unifying perspective: Clear derivation that places softmax and linear attention in one framework, making the “first-order approximation” view explicit and intuitive.

Empirical support: Multiple datasets and two model scales; scaling experiments (longer context, larger model) show the normed variant tracks softmax closely; thorough ablations (Taylor order, gating vs norms, linear-attention variants)

Didactic exposition: The RNN view and Kronecker-product formulation help reason about “higher-order interactions” and why linear attention underperforms.

**Claims And Evidence:**

Yes

**Claims Explanation:**

Overall, the paper presents substantial analysis and derivations; although some assumptions are occasionally loose, the conclusions appear largely sound.

**Requested Changes:**

Task breadth is limited: Results focus on next-token LM loss curves; there are no standard perplexity benchmarks, downstream tasks, in-context learning evaluations, or long-range reasoning tasks to confirm functional parity with softmax beyond train/test loss.

Denominator “norm” claim needs stress tests: The observation that a simple vector norm suffices is shown on a few setups; stability/robustness across wider context lengths (e.g., 32K+), architectures, and domains (vision, speech) remains unclear. The gate variant’s instability is acknowledged but not deeply analyzed.

For journal papers, it’s best to use vector graphics for figures.

---

> ### Author Response · Authors · 2025-09-19
> **Author Response**
>
> Thank you for taking the time to thoroughly review our paper. We provide the train and test log loss curves to give a baseline comparison between softmax and the proposed model architectures. We agree that adding additional benchmarks can strengthen and provide more evidence for the proposed equivalences. However, due to computational constraints (8x A100 80GB GPUs), we trained "medium-sized" models compared to the current SOTA LLMs, which would require many more computational resources. Using these modest models sizes, most benchmarks would not be performant. Even so, we could provide a needle-in-a-haystack test to show that our proposed methods can perform as well as softmax outside of the standard next token prediction.
>
> The reviewer notes that the norm should be evaluated on tasks other than causal language modeling. Given the constraints on compute, this analysis cannot be completed within a reasonable time frame. Rather than empirically show this generalization, we have attempted to add qualifying language to the claim about the denominator acting as a norm, and have added some text that acknowledges this specifically as a limitation. From this perspective, the strength of the claim is reduced, which we hope is an acceptable middle ground for the requested change.
>
> We agree that some of the diagrams are difficult to see due to not being vector graphics. We have added a section in the appendix that expands these figures for better viewing purposes.

---

### Decision · Action_Editor_ur96 · 2025-09-28

**Recommendation:** Accept as is

**Audience:**

Yes

**Audience Explanation:**

All three reviewers confirmed this. The paper addresses a fundamental question about why softmax attention is effective, provides a unifying theoretical framework connecting transformers and RNNs, and offers insights relevant to attention mechanism researchers and those working on state-space models.

**Claims And Evidence:**

Yes

**Claims Explanation:**

All three reviewers confirmed the claims are supported by accurate, convincing and clear evidence. The mathematical derivations are rigorous and the empirical validation is extensive across multiple datasets and model scales.